# How the Nonwoven Polymer Volume Microstructure Is Transformed under Tension in an Aqueous Environment

**DOI:** 10.3390/polym14173526

**Published:** 2022-08-27

**Authors:** Elena Khramtsova, Egor Morokov, Christina Antipova, Sergei Krasheninnikov, Ksenia Lukanina, Timofei Grigoriev

**Affiliations:** 1Emanuel Institute of Biochemical Physics, Russian Academy of Sciences, 119334 Moscow, Russia; 2National Research Center ”Kurchatov Institute“, Department of Nanobiomaterials and Structures, 123182 Moscow, Russia; 3Moscow Institute of Physics and Technology, National Research University, 141701 Moscow, Russia

**Keywords:** electrospun polymers, mechanical behavior, ultrasonic imaging, microstructure, poly(L-lactide), volume imaging, tissue engineering

## Abstract

The fibrous porous structure of polymers can mimic the extracellular matrix of the native tissue, therefore such polymers have a good potential for use in regenerative medicine. Organs and tissues within the body exhibit different mechanical properties depending on their functionality, thus artificial scaffolds should have mechanical behaviors similar to the extracellular matrix in conditions like living organisms, primarily in aqueous media. Several methods have been investigated in aquatic environments, including noninvasive techniques based on ultrasonic focused beams for biological objectives. In this study we explored the tensile behavior of poly(L-lactide) nonwoven polymer scaffolds using high-frequency ultrasound microscopy combined with a horizontal testing machine, which provided a visualization of the reorganization and transformation of the dynamic volume microstructure. The mechanisms of unwinding, elongation, orientation, and deformation of polymer fibers under uniaxial tension were revealed. We observed an association between the lined plastic deformation from 100 to 400% and the formation of multiple necks in the fibers, which caused stress relaxation and significant rarefaction of the fibrous microstructure. It was shown that both peaks on the stress–strain curve corresponded to the microstructure of aligned fibers in terms of initial diameter and thinning fibers. We discuss the possible influence of these microstructure transformations on cell behavior.

## 1. Introduction

Behavior of artificial scaffolds under load mimicking natural processes in the living organism is the most important parameter during the formation and selection of implants. Artificial materials with a high specific surface area and high porosity, which together provide effective attachment of cells and delivery of nutrients, have great potential for regeneration of the connective tissue of internal organs, skin, vascular system, and bones [1,2,3,4,5]. The extracellular matrix is individual for each tissue and has specific properties that form a unique tissue-specific environment that provides not only vital activity but also cell differentiation. Various tissues radically differ in morphology, matrix composition, and density [6]. Some tissues (large nerve fibers, heart, blood vessels, tendons, etc.) have poor physiological regeneration and restore via scar formation, resulting in loss of functionality. To fix and to improve tissue recreation and regeneration, aligned or partially orientated structures could be directly applied to cells’ development and differentiation, according to physiological regeneration processes [1]. These materials could be formed by electrospinning methods. The main parameters of nonwoven electrospun materials include polymer composition, fiber diameter, degree of porosity, packing density, and orientation, all of which significantly affect the mechanical properties of the material [7,8,9,10,11] and the future success of scaffold implantation processes [12,13,14,15,16,17,18,19].

Nonwoven artificial scaffolds that mimic the extracellular matrix in tissues should provide mechanical function at three hierarchical levels, i.e.: macroscopic (eg. whole nonwoven mats); microscopic, estimating interaction of threads and their orientation; and the molecular structure of individual fibers [20]. The mechanical behavior of scaffolds has a significant effect on the process, primarily in the post-implantation response to the material in vivo [17,21,22]. Therefore, it is necessary to assess the contribution to the behavior of cells made by the geometry and properties of individual fibers [23,24], and their random [18,25,26,27,28] or aligned mesh [15,29,30,31], with the precisely controlled orientation of fibers [19]. 

Frequent publications have been devoted to cell adhesion and proliferation in the scaffold of various fiber packages. The ability to migrate into depth depends directly on the spatial characteristics of the material [12,14,16]. For fibroblasts, it is necessary to control both the diameter of fibers and the distance between them (pore size); thin densely packed fibers could impede effective fibroblast migration [27,28]. Another way to increase survival, adhesion, and differentiation of cells is the inducement of the extracellular matrix of the target organ into nonwoven scaffolds, prompting the biochemical signals in the scaffolds [32]. However, cell behavior is not consistent if the microstructure of the nonwoven scaffold changes over time. In particular, when a fiber is strained, the cell senses that this fiber is stiffer [12,16,24]. Hodge and Qint [22] investigated cell proliferation in vessel scaffolds during low cyclic mechanical stretching, and demonstrated a dramatic enhancement of collagen production in human dermal fibroblasts compared with static vessels’ scaffold. For the first time, this approach was applied for tissue engineering of heart valves [33], where an increase in collagen concentration, an increase in vimentin expression, and a more homogenous cell distribution were observed.

Since there is a close correlation between cell activity and scaffolds characteristics, the creation of artificial matrixes with a defined microstructure and predictable properties is an important issue in tissue engineering. Internationally, novel approaches to the formation of nonwoven scaffolds with controlled fiber orientation have been developed [11,19]. Qin et al. [19] described the correlation between cell migration velocity and the angle between fibers, the fiber diameter, and the distance between fibers. Recently attention hs focused on the dependence of the mechanical properties of nonwoven mats on fiber orientation, fiber diameter, porosity, fiber–fiber bonding, etc. [10,23,34,35,36]. Despite active research in this field, a study of the mechanical behavior of electrospun materials under stress remains relevant owing to the current lack of experimental data, primarily data on the evolution of microstructure in the scaffold volume. The study of mechanical behavior is a complex task requiring effective experimental approaches to the observation of changes in micromechanical dynamics. Several experimental approaches [7,9,20,37,38,39,40,41] have been applied for in situ monitoring of failure processes in nonwoven materials. Equipment based on optical and electron microscopy made it possible to observe the deformation of nonwoven scaffold only from the surface [7,9,20,37,38]. It is necessary to apply methods to image microstructure evolution in the scaffold volume. However, only two methods have provided resolution of tens of microns or less in the polymer electrospun materials, based on probing by ultrasound [42,43] or X-ray radiation including synchrotron radiation [39,40,41,44]. Each method has its own limitations based on the mechanism of contrast defining the area of application. Usually, X-ray methods require a relatively long exposure time, which may lead to stress relaxation in the specimen volume. Moreover, when examining fibrous objects for biomedical purposes, it is important to assess the mechanical behavior in conditions similar to the natural conditions in a living organism; the aqueous environment is of primary relevance. The above-mentioned methods (optical and electron microscopy, X-ray techniques) have generally been applied for in situ monitoring in air only, but the mechanical properties of nonwoven matrix measured in air and in water may have significant differences [35,45,46]. The optimal method of studying objects in the aqueous environment is ultrasonic microscopy, with water immersion being a prerequisite for the experiment. Moreover, ultrasound methods are noninvasive and have a good natural contrast with cells, which gives a perspective for the investigation of cell proliferation through the volume of fibrous scaffolds.

Herein, for the first time, ultrasonic microscopy was applied to visualize and analyze the mechanical behavior of randomly oriented poly(L-lactide acid) (PLLA) electrospun scaffolds under uniaxial tension. Polylactide is a biodegradable and cell-compatible polymer widely used for biomedical applications. PLLA-based biomaterials could be fabricated from nano- and microfibers into a variety of structures and different fixation devices in orthopedics and dentistry (pins, screws, etc.), thus, the study of polymer products is of both practical and fundamental interest. Experimental data from PLLA fibers [17,47] as well as polycaprolactone [20], polystyrene [8,9], polypropylene [39], polyhydroxybutyrate [43,48,49] polyamide [36,50], have been applied to develop a theoretical representation of the micromechanics of fibrous materials [8,9,50,51,52]. Nonwoven polymer mats are characterized by different mechanical properties (strength, elongation at break, elastic modulus, etc.) that depend on many factors: nature of polymer, fiber diameter, porosity, fiber orientation, quality of fiber–fiber connection, etc. Estimation and prediction of the influence of all factors on the mechanical behavior are among the leading issues in polymer science, including descriptions of transformations appearing in the volume of electrospun materials under loads.

The motivation of the current work was the lack of knowledge about the correlation between mechanical behavior and the evolution of volume microstructure in nonwoven scaffolds under tension in an aqueous environment that mimics a living organism. To highlight the mechanisms of microstructural transformations from single fibers to fibrous mesh, the in situ ultrasound monitoring of microstructure evolution was carried out. We observed the micro- and macrostructural transformations in the volume of scaffolds at different stages of deformation. The ultrasonic data of the volume microstructure was supplemented with optical microphotographs and load curves of the tested specimens. We also discuss possible changes in cell proliferation arising from the effects of microstructure transformation in the scaffold volume.

## 2. Materials and Methods

### 2.1. Specimen Preparation

The fibrous material was obtained by electrospinning 9 wt.% solution of poly(L-lactide acid) (PLLA 4032D, NatureWorks LLC, Minnetonka, USA, molecular weight is 200 kDa) dissolved in the mixed solvent (90% chloroform and 10% ethanol). The solution was homogenized on a stirrer for 12 h. A rotating drum with a speed of 40 rpm and a diameter of 12.5 ± 0.5 cm was used as a precipitating electrode. The voltage between electrodes was kept in the range of 13 ± 3 kV. More detail of the electrospinning process has been described previously, for example [53]. After being obtained, the material was dried in a ThermoFisher vacuum oven to remove residual solvent. The interelectrode distance was 20 ± 5 cm. The degree of crystallinity of the individual fiberswas 20–40%. The swelling of the fibers in water was no more than 2%. Testing samples were cut from the centers of the finished mats, avoiding the edge zones, for ultrasonic, optical, and mechanical studies.

The pure fibrous PLLA material was characterized by layer thickness, packing density, and average fiber diameter. The thickness of the material was measured with a SHAN 123758 micrometer. The mean value was 400 ± 10 μm. A Carl Zeiss Axio Imager.M2m optical microscope was used to examine fiber diameters in the material.

### 2.2. Mechanical Tests

The mechanical properties of the fibrous material were studied under uniaxial tension on an Instron 5965 testing system. The material was cut into pieces along the generatrix of the rotating drum. The samples each had a width of 5 mm and a length of 50 mm. Before tests, all samples were soaked in distilled water for 48 h. To facilitate soaking, the samples were preliminarily immersed in ethanol for 5 min. The working part of the samples was 10 ± 1 mm. The tests were conducted at a speed of 25.0 ± 0.5 mm/min. The temperature was equal to 22 ± 2 °C and the humidity—50 ± 10%. For statistical analysis, ten moist samples were taken.

### 2.3. Acoustic Microscopy

The study of microstructures in the volume of nonwoven scaffolds was carried out with the Scanning Impulse Acoustic Microscope (SIAM-2017) developed at the Emanuel Institute of Biochemical Physics, Russian Academy of Sciences. Previously, the possibility was demonstrated in the works of applying high-frequency ultrasound for visualization of the volume microstructure in nonwoven materials [42,43]. Here we used a lens with a working frequency of 200 MHz (wavelength is 7.5 µm) and a half-angle aperture of 11° that gave a focal spot diameter of 24 µm through a focal depth of 400 µm. The in situ visualization experiments were carried out in distilled water. The combination of the microscope and a horizontal mechanical testing machine (Figure 1) made it possible to observe dynamically the changes of microstructure. We used the conventional step-by-step methodology, a visualization of the same nonwoven specimen at different stages of loading. The detailed methodology of the experiments has been described previously [43]. Briefly, in the analysis of load curves obtained from mechanical tests, we selected strains, in which the internal arrangement of fibers could be changed. These elongations were set on the horizontal testing machine and maintained during the ultrasound investigations. Thus, ultrasound data included tomographic images of the volume microstructure from the same sample at different deformations. The results of ultrasound scanning were presented as sets of horizontal cross-sections in the volume of the specimen (C-scans), in which the sequences of events (appearance and growth of damages) were revealed. The surfaces of the nonwoven materials were visualized with a Leica DMLM (Germany) optical microscope and a Nikon digital camera. Ultrasound and optical monitoring were conducted on samples of 10 mm × 5 mm standard geometry for mechanical testing.

## 3. Results

### 3.1. Nonwoven Sample Characteristics

Using SEM images of the microstructures of nonwoven PLLA test specimens (Figure 2a), we observed a spread of fiber diameter that varied from 2.7 to 6.4 µm with an average value of 5.0 µm (Figure 2a). The packing density for PLLA material was equal to 9 ± 2%.

The load–deformation curves from the mechanical test data showed the classical three regions: (1) initial region (elastic zone); (2) nearly linear region; and (3) failure region (Figure 2b). It can be observed in the load diagram that the point of transition from one section to another appeared in the same deformation for all specimens. The limit of elastic deformation was about 50% of scaffold strain when the first extremum point was observed. The results indicated that scaffolds exhibited a spread of Young’s modulus from 68.8 to 128 MPa. Plastic deformation showed significant elongation of the material with maximum stress value (second peak) at strain of about 400%. The stress value for each specimen was associated with the average fiber diameter in the tested nonwoven material, and a reducing or rising of the diameters affected this [8]. In our case, the ratio between the quantity of thick and thin fibers in each sample could probably influence the stress values. The values spread over the stress axis do not influence the mechanical behavior of these types of nonwoven material in general. Thus, for visualization of microstructure evolution, we considered the typical deformation stages under tension—elastic strain of 0, 30, and 50% and plastic deformations of 250, 400, and 550%.

### 3.2. Imaging of Individual Fibers under Tension

To describe the transformation of microstructure in the volume of nonwoven mats imaged by ultrasound microscopy, we previously obtained the results for single fibers. Ultrasonic scans are formed as raster gray-scaled images where the brightness of the pixel is determined by the amplitude of the signal reflected or scattered at each fiber. The diameter of the fibers is much less than the focal spot of the probing ultrasound beam (5.0 µm to 24 µm), therefore all fibers seen in the images (C-scans) are displayed as elements with diameters equal to the focal spot, but with different brightness. The differences become visible during mechanical testing. Because fiber diameter is decreased under tension, the amplitude of back-scattered ultrasound signals is reduced too, resulting in a drop of the brightness of the imaged fibers. This means that fibers or local regions of fibers of lower diameter are displayed in the ultrasonic image as dim elements. In Figure 3 the ultrasound and optical images of individual elongated fibers are demonstrated. The original diameter of each fiber was 5 µm, and the neck was half this thickness. It was observed that the appearance of fiber necks, which usually occurs in amorphous regions [54], decreased the ultrasonic back-scattered signal (a drop of 8 dB) and the brightness of the corresponding regions in the fiber. Thus, ultrasound images of microstructure in the volume of nonwoven test scaffold included elements of different brightness corresponding to fibers with thinned and original diameters. We estimated the quantity of stretched and initial fibers in the matrix volume by assessing changes in fiber brightness. Because each backscattered echo generally corresponds to one fiber, we also estimated the fiber volume fraction in the vertical cross-sections (B-scans) of the nonwoven samples under different strains.

### 3.3. Evolution of Microstructure in the Volume of Scaffolds under Tension

To investigate the sequence of microstructure transformation we chose six points on the stress–strain curved, and the tested nonwoven sample was scanned at the selected deformations (Figure 4). A fixed depth in the nonwoven materials (at the middle of the specimen thickness) was chosen for ultrasound imaging. In the ultrasonic images (C-scans) we visualized the fibers in a finite volume of 50 µm thick in the scaffold’s bulk to provide the estimation of individual fibers, their diameter, and orientation. The ultrasonic data were complemented by optical microphotographs obtained at the same deformations; reflected light microscopy imaged the integral structure of the test sample seen from the top surface. It should be noted that acoustic microscopy data included the full 3D tomographic model of the whole thickness, and middle thickness was manually selected for each deformation. The full ultrasound data was used for the fiber volume fraction calculation (Section 3.4).

The intact structure of the PLLA nonwoven mats (Figure 4a) contained curved and aligned fibers. The images display a random mesh of fibers. In stage 1 (Figure 4b), the twisted and hooked fibers were unwound and oriented along the tensile axis. The areas of untouched microstructure connected by straightened fibers are seen in the images of 30% deformation. Several bundles of fibers were oriented along the tension axis. At the first extremum point (Figure 4c), a major part of the fibers were unwound, and stress was distributed over them. In the C-scans, all fibers had the same brightness markingd the integrity of the original geometry of the fibers and the absence of necks on them. The packing density of the fibers was reduced, as shown in the ultrasound image (Figure 4c). A small stress relaxation immediately after the peak at 50% deformation was detected in all test specimens (Figure 2b). The drops of stress–strain curves could indicate the formation of multiple necks in the fibers. The second linear section of the load diagram corresponds to the continuous elongation of most of the fibers in the volume (Figure 4d). The C-scan showed fibers of different brightness, indicating a reduction in the diameter of the fibers (deformation of 250%). Many twisted and hooked fibers continued to be seen in optics (Figure 4d). As strain increased, deformed thin fibers were frequently observed by ultrasound, until the majority of the thinned fibers reached their elongation limit. This point corresponded to the second extremum and the value of maximal stress at the strain of 400%. In images (Figure 4e) we observed the aligned fibers, which are more clearly visible in the ultrasound image of the narrow volume of the scaffold. The fiber packing became denser compared to the microstructure at 250% strain, (Figure 4d), however, the low brightness of fibers indicated a decrease of their diameter. After reaching the elongation limit of thinned fibers, successive fiber breaks appeared. Nevertheless, in the volume, there remained the fibers that were most curved initially. They held the load, and lengthened and broke one by one (Section 3 of the load curve). We also observed orthogonal and hooked elements that influenced the tested sample’s durability. A rarefied microstructure was seen in optical and ultrasonic images (Figure 4f) of material at stretching of 550%.

### 3.4. Estimation of Fiber Quantity in Vertical Cross-Section at Different Deformations

To calculate the fiber volume fraction at various elongations of PLLA nonwoven scaffolds, we took B-scans of the samples and manually counted the fibers. The signals’ shapes and phases were taken into account during estimations. The number of fibers was evaluated in a limited square of 0.1 mm × 1 mm within the vertical section of the sample, orthogonal to the tension axis. The calculation data is presented in Table 1. To clearly display the change in scaffold density, all images of vertical sections (B-scans) were converted to a black-and-white color scheme (Figure 5). The amplitudes of the echo signals were taken modulo, to provide the best contrast, losing the phase transformations of the signals used for manual calculation of fiber quantity.

Looking at vertical cross-sections of the nonwoven materials under elastic stretching ε from 0 to 50% (B-scans, Figure 5), we observed the dense packaging that decreased to 70 fibers per 0.1 mm^2^. The initial elongation of scaffolds corresponded to the elastic region of the stress–strain curve, retaining the original diameters and structure of fibers, though the orientation of hooked and twisted fibers changed. The geometry of the tested sample deformed, the width in the central part reduced, as did total thickness (Figure 5, B- and C-scans at ε = 0, 30, and 50%). The diagonal fibers were imaged in a C-scan obtained at the strain of 50%. Stress after the first extremum point was relaxed by multiple local neck formation in the volume of the sample. In the vertical section, at a deformation of 250% (B-scan, Figure 5), could be observed an increase in the distance between the fibers and a decrease in the packing density of the test sample to 40 fibers per 0.1 mm^2^. At the same time, the average thickness of the scaffold increased to 450 µm (Table 1). When the sample was stretched, some of the fibers were torn and other fibers became thinner. Around the second extremum point, the fibers in the volume reorganized into a dense bundle aligned along the tensile axis. The peak strength at deformation of 400% was achieved by fibers of reduced diameter at maximal elongation. The fiber volume fraction increased compared to the previous point of deformation (ε = 250%), and amounted to 54 fibers per 0.1 mm^2^ (Table 1), but the sample had thinned (Figure 5). The nonwoven sample lost its satiability after the instant breaks of numerous thinned fibers, nevertheless, in the volume, there remained the fibers that were the most curved initially. They held the load and one by one lengthened and broke. We also observed orthogonal and hooked elements that influenced the tested sample’s durability. The fiber volume fraction at a strain of 550% was equal to 40 fibers per 0.1 mm^2^ and an average thickness of 140 µm.

## 4. Discussion

In the work, mechanical behavior of scaffolds that mimics extracellular structure in tissues was visualized and considered. For the first time, the transformation of the microstructure of nonwoven materials in environments similar to living organisms, namely in the aqueous environment, has been depicted. We used a scanning acoustic microscope with short-pulse generation and a long-focus acoustic lens at an working frequency of 200 MHz, combined with a horizontal mechanical testing machine. This combination resolved the problem of in situ imaging of changes of microstructure in the volume of nonwoven scaffolds under tension. PLLA electrospun materials were taken as test samples, due to their biocompatibility and wide medical application. The mechanical behavior of single fibers and meshes was assessed; we found correlation between the molecular and supramolecular structure of the polymer. The elongation of PLLA fibers was accompanied by the appearance of necks in amorphous regions of the polymer. PLLA is a semi-crystalline polymer with crystallinity up to 45% [55,56], and necks were intensively formed in the elongated nonwoven structures (Figure 3). Intensive necking was observed in the volume of tested mats under in situ ultrasound monitoring. The C-scans (Figure 4) depicted a thin layer of 50 µm at the middle depth of the nonwoven specimen. This means that the scans showed only a limited number of fibers compared with the optic microphotographs, which presented all fibers visible from the surface of the specimen. Thus the mechanisms of unwinding (Figure 4b), orientation (Figure 4c), and elongation (Figure 4d,e) of individual polymer fibers were revealed on C-scans. Based on the data obtained, features of the tensile behavior of randomly orientated PLLA meshes can be formulated as follows:under elastic deformation, the unwound fibers retain their original diameter and structure, and are oriented along the tensile axis (Figure 4a,b);when most of the stretched fibers of the original diameter unbend (Figure 4c), the nonwoven scaffold reaches the first maximum on the stress–strain curve. The elastic region ends;the long-lined elongation of scaffolds starts from the drop in stress caused by multiple necking in the fibers (Figure 4d). The spun material becomes rarefied (Figure 5, ε = 250%), fiber diameters decrease, interfiber distance grows (Table 1);the maximal load (second peak) on the scaffolds arises when the most of the thinned fibers are aligned over the axis (Figure 4e). The fibers undergo deformation close to tensile strength. The samples lost their satiability after the instant breaks of the numerous thinned fibers;the last stage of elongation corresponds to lengthening and breaking of most of the initially curved fibers. They were imaged with the same brightness (Figure 4f) as fibers of the initial structure (Figure 4a), while the previously elongated fibers looked darker (Figure 4f). The fibers held the load, elongated and broke one by one. Therefore, the load curves of the specimens each showed a long ending.

Generally, the tension process in the PLLA nonwoven scaffolds with random mesh could be divided into three stages (Figure 2b). The initial elongation of scaffolds corresponding to the elastic region of the stress–strain curve (Section 1) is usually associated with the deformation of collagen matrix in tissues [57]. Stress relaxation in scaffolds before reaching turning point (the first extremum) retains the initial structure of fibers, though the orientation of hooked and twisted fibers can change. Low tension load decreases the angle at the points of fiber junctions (Figure 4b,c) and reduces the vertical interfiber distance (B-scans in Figure 5). In the presence of cells in the scaffold, these transformations in microstructure could affect cell activity. Primarily, cells orient themselves along the aligned fibers [15,30], and a decrease in the distance between the fibers can negatively affect cell migration over the scaffold depth [16]. Thus, cell proliferation will predominate in the plane of the stretched fibers. Along with this, cells sense the stretched fibers as being stiffer than the initial material, which may improve cells’ adhesion to the fiber surface [16,24]. Probably, such effects had been observed in works [22,33] where low-level cyclic loads within elastic deformations improved cells’ proliferation, such as collagen fibers’ formation in artificial scaffolds.

The next stage of tension (Section 2 of the stress–strain curve in Figure 2b) is initiated with the ultimate stretching of the majority of fibers. The nonwoven sample elongated with the thinning fibers, and necks appeared in different amorphous regions and joined together under tension (Figure 3). A twofold decrease in fiber diameter was observed. It was shown that stress relaxation caused by necking resulted in an increase in the distance between fibers (Figure 5, ε = 250%) and reduced the packing density of the tested specimen. Thus, exceeding the limit of elastic deformation leads to a double effect of the reduction of fiber diameters and the growth of porosity in nonwoven materials. Such changes could affect cell adhesion, as well as reduce cells’ migration rates. Reorganization of the thinned nonwoven fibers during tension and their compaction results in a structure similar to aligned electrospun materials. Gaharwar et al. [30] demonstrated the same necking behavior of fibers in mechanically deformed poly(glycerol sebacate)–poly(ε-caprolactone) scaffolds. The deformed fibers showed uniform smooth surface morphology. It is difficult to imagine cell behavior after all stresses on scaffolds, especially in deformations that are unlike the stresses of natural tissues, therefore, the effects of cells caused bysignificant elongations of nonwoven materials are seldom considered.

Thus, in this work we have demonstrated the evolution of the microstructure in the volume of nonwoven mats. Changes in stiffness, porosity, fiber orientation, packaging, and necking were described, and correlated with the load curve. Understanding scaffold properties and their behavior could give insights into the regulation of cell differentiation through mechanotransduction. The question of the behavior of tissue engineering constructs (scaffold + cells) under loads in conditions close to living organisms remains an area of interest.

## 5. Conclusions

In this work we studied the mechanical behavior of PLLA nonwoven scaffolds under uniaxial tension and related their properties to changes of microstructure in the volume. The study was conducted with high-frequency ultrasound microscope combined with a horizontal testing machine, to provide in situ monitoring of the dynamic evolution of the microstructure. Data was obtained for specimens in aqueous environment, under conditions close to living organisms, which plays a significant role in tissue engineering, primarily for predicting cell behavior under stress. We identified and described several mechanisms responsible for the extremum points on the stress–strain curve. Both peaks in the load graphic correlated with reorientation of fibers along the load axis. It was shown that necking is a reason for significant stress relaxation, which results in rarefaction of the fibrous microstructure.

## Figures and Tables

**Figure 1 polymers-14-03526-f001:**
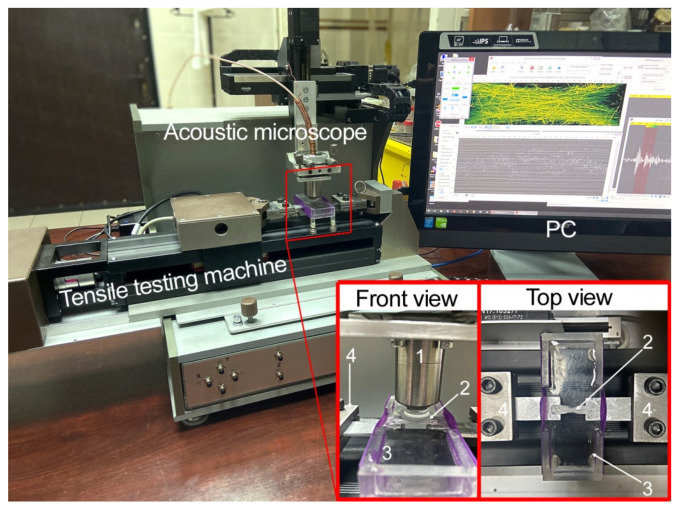
Experimental setup—acoustic microscope combined with horizontal testing machine. The microscope and the tensile machine were controlled form one PC. The specimen (2) was glued to the metallic holders that were placed into the clamps (4) and passed through the elastic side of the water tank (3). The acoustic lens (1) was immersed in water and positioned above the specimen surface, while its focus was localized in the volume of the sample.

**Figure 2 polymers-14-03526-f002:**
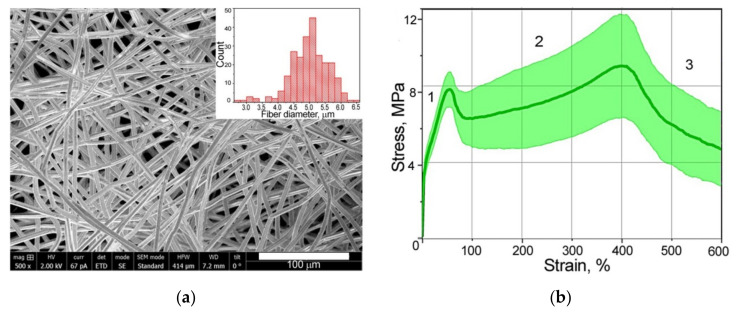
(**a**) SEM image of PLLA materials; (**b**) histogram of fiber diameter spread and averaged strain–stress curves of tested samples, showing statistical spread.

**Figure 3 polymers-14-03526-f003:**
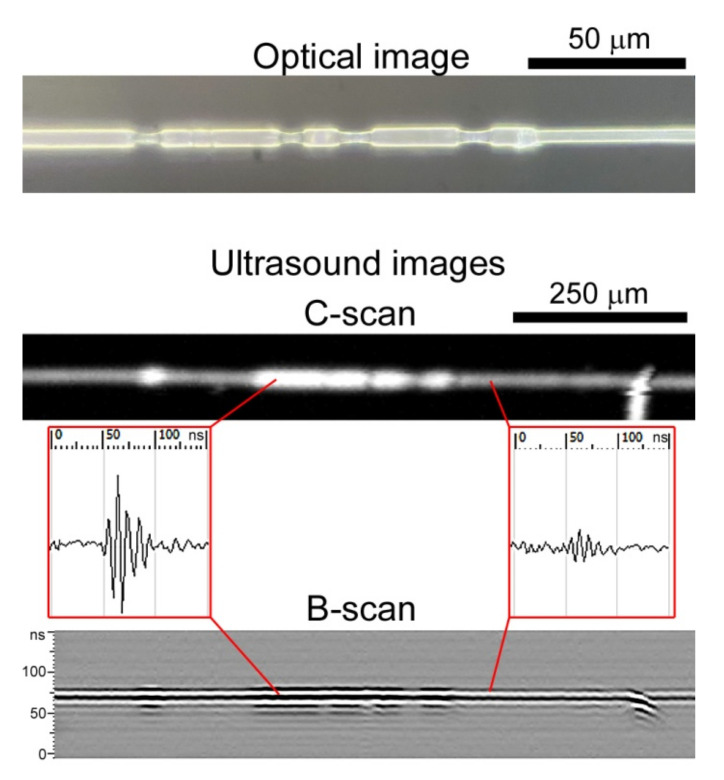
Visualization of a single stretched PLLA fiber using light microscopy and acoustic microscopy. The original diameter of the fiber was 5 µm. The thinned regions of the fiber generate a backscattered signal of lower amplitude in comparison with the original diameter, providing different observations of such regions in ultrasound images. Thin elongated fiber regions are dimly displayed. The field of view in the C-scan is 120 µm × 1000 µm (60 × 200 DPI). The B-scan was obtained from the fiber center. The microphotograph was obtained at the magnification of ×1000.

**Figure 4 polymers-14-03526-f004:**
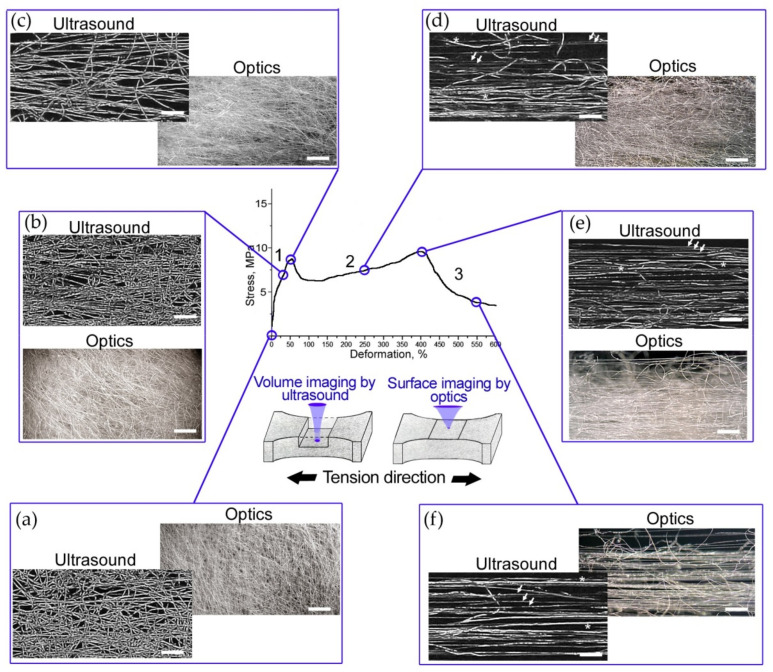
Step-by-step imaging of the test PLLA sample using light and acoustic microscopy at deformations of (**a**) 0, (**b**) 30, (**c**) 50, (**d**) 250, (**e**) 400, and (**f**) 550%. Microphotographs were obtained at a magnification of ×50. They show integral structures seen from the top surface of the nonwoven scaffold. Ultrasound images were obtained at the central depth of the test sample. They depict only the fibers that are part of a narrow volume with a thickness of 50 µm in the bulk. (**a**–**c**) Both types of images show fibers of the original diameter only at strain up to 50%. The ultrasound images (**d**–**f**) show an example of thinned elongated fibers, and asterisks indicate fibers of initial diameters. The scale bar is 500 µm.

**Figure 5 polymers-14-03526-f005:**
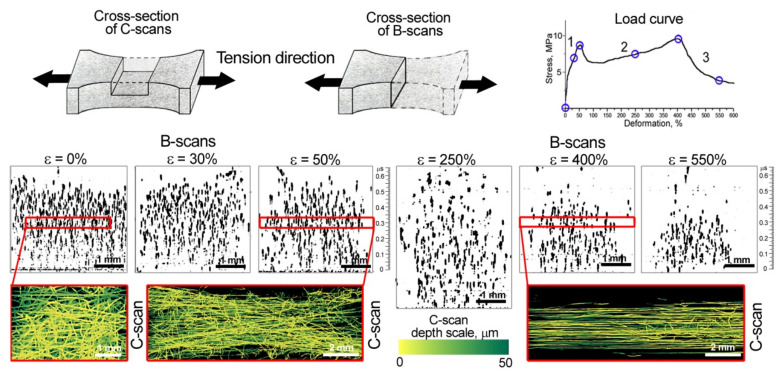
Ultrasound imaging of fiber packing density at different stages of deformation ε. B-scans were obtained from the central regions of the tested nonwoven sample. To display the vertical cross-sections (B-scans), the signal amplitudes were taken modulo. The color C-scans depict the distribution of fibers over a thickness of 50 µm at the middle depth of the scaffold, under a load corresponded to two peak points at elongations of 50 and 400%.

**Table 1 polymers-14-03526-t001:** Average fiber-packing density data for various deformations of scaffolds, according to ultrasound.

Deformation, %	0	30	50	250	400	550
Fiber volume fraction in the central part ± 10, ×0.1 mm^−2^	145	94	67	40	54	40
Average thickness of scaffold, µm	380	350	320	450	250	140
Average width of scaffold, mm	5	3.7	3.3	3.3	3	2.4

## Data Availability

Not applicable.

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
