# Peer review of "How the Nonwoven Polymer Volume Microstructure Is Transformed under Tension in an Aqueous Environment"

_polymers, 2022, doi:10.3390/polym14173526_

Round 1

Reviewer 1 Report

The fibrous porous structure of polymers have a good potential in regenerative medicine application. This work focus on the transformation characteristics of microstructure under tension in an aqueous environment.  There are some work should be improved before is accepted for publication:

1. there are some formation errors in paper should corrected;

2. what is the effect of nanofiber diameter on the mechanical properties?

3. what is the effect of nanofiber distribution on the mechanical properties?

Author Response

Response to Reviewer 1 Comments

The fibrous porous structure of polymers have a good potential in regenerative medicine application. This work focus on the transformation characteristics of microstructure under tension in an aqueous environment.  There are some work should be improved before is accepted for publication:

  1. there are some formation errors in paper should corrected;

Response 1: The manuscript has been checked for spelling and grammar

  1. what is the effect of nanofiber diameter on the mechanical properties?

Response 2: Typically, a reduction in fiber diameter affects the maximum load that can be applied. The load will decrease. On the other hand, electrospun materials with nanofibers have a denser packing compared to samples with micron-sized fibers. Both aspects could influence on the mechanical properties; however, the behavior of the materials should be staying the same. Three sections on the load curve will be visible: elastic region, line region with peak stress and section of drop of stress.

  1. what is the effect of nanofiber distribution on the mechanical properties?

Response 3: The distribution and orientation of the fibers significant influence of elastic behavior of the electrospun materials. Aligned fibers have lower elongation but higher maximum load compared to random fiber distribution.

Reviewer 2 Report

The novelty of this work is good. Here are my comments:

1. The authors should provide the scheme for the materials used in this study.

2. The reviewer suggests the authors improve the quality of the figures.

3. How about the thermal stability of these materials?

4. What is the molecular weight of PLLA 4032D in this study?

5. The conclusion part is too long.

Author Response

Response to Reviewer 2 Comments

The novelty of this work is good. Here are my comments:

  1. The authors should provide the scheme for the materials used in this study.

Response 1: Nowadays, the production of electrospun materials is a common technology, and here we have only described the formation parameters of this PLLA nonwoven. In section 2.1 we introduce the sentence: “More detailed electrospinning process was described earlier, for example [Lukanina, K., Grigor’ev, T.; Tenchurin, T. ; Shepelev A.; Chvalun S.  Nonwoven Materials Produced by Electrospinning for Modern Medical Technologies (Review). Fibre Chem, 2017, 49, 205–216]”

  1. The reviewer suggests the authors improve the quality of the figures.

Response 2: The resolution of figures has been improved.

  1. How about the thermal stability of these materials?

Response 3: Of course the polymer is degradable and unstable under heating. According to TGA PLLA 4032D is thermally stable upon 250 ºC in oxygen atmosphere. Thus, during the experiment, the material was thermally stable at room temperature in organic solvent. More interesting is to investigate the influence of hydrothermal aging and resorbtion of the polymer on mechanical behavior and the nonwoven volume microstructure. It will be analyzed in our future studies.

  1. What is the molecular weight of PLLA 4032D in this study?

Response 4: We added the data of molecular weight in the text. The value is 200 kDa

  1. The conclusion part is too long

Response 5: Conclusion has been rewritten.

Reviewer 3 Report

The paper is devoted for fibrous materials investigations by mechanical tests and acoustic microscopy. The topic is generally interesting, however the paper contain unexplained places (below) and need major revisions.

The aim of the paper should be more clearly explained.

The discussion part should be rewritten in more logic way.

Figure 4 should be more discussed.

Conclusions should be rewritten in more informative way.

All typos should be corrected, for example lines 301, 307 ’’mm2’’. Figures should be cited inside the sentence, please see for example line 280.

English need minor revisions.

Author Response

Response to Reviewer 3 Comments

The paper is devoted for fibrous materials investigations by mechanical tests and acoustic microscopy. The topic is generally interesting, however the paper contain unexplained places (below) and need major revisions.

  1. The aim of the paper should be more clearly explained.

Response 1: We add our motivation in the end of Introduction

  1. The discussion part should be rewritten in more logic way.

Response 2: Discussion has been rewritten

  1. Figure 4 should be more discussed.

Response 3: We add the description of Figure 4 in discussion

  1. Conclusions should be rewritten in more informative way.

Response 4: Conclusion has been rewritten.

  1. All typos should be corrected, for example lines 301, 307 ’’mm2’’. Figures should be cited inside the sentence, please see for example line 280.

Response 5: The text has been improved.

  1. English need minor revisions.

Response 6: The manuscript has been checked for spelling and grammar

Round 2

Reviewer 2 Report

This manuscript should be published in this journal.

Reviewer 3 Report

Authors make proper corrections according to reviewer remarks and I suggest to publish the paper as it is.